# Phonon heat transport in cavity-mediated optomechanical nanoresonators

Cheng Yang[1], Xinrui Wei[1], Jiteng Sheng [1,2]✉ & Haibin Wu [1,2]✉

The understanding of heat transport in nonequilibrium thermodynamics is an important research frontier, which is crucial for implementing novel thermodynamic devices, such as heat engines and refrigerators. The convection, conduction, and radiation are the well-known basic ways to transfer thermal energy. Here, we demonstrate a different mechanism of phonon heat transport between two spatially separated nanomechanical resonators coupled by the cavity-enhanced long-range interactions. The single trajectory for thermalization and non-equilibrium dynamics is monitored in real-time. In the strong coupling regime, the instant heat flux spontaneously oscillates back and forth in the nonequilibrium steady states. The universal bound on the precision of nonequilibrium steady-state heat flux, i.e. the thermodynamic uncertainty relation, is verified in such a temperature gradient driven far-off equilibrium system. Our results give more insight into the heat transfer with nanomechanical oscillators, and provide a playground for testing fundamental theories in non-equilibrium thermodynamics.

[1] State Key Laboratory of Precision Spectroscopy, East China Normal University, Shanghai 200062, China. [2] Collaborative Innovation Center of Extreme Optics, Shanxi University, Taiyuan 030006, China. ✉email: jtsheng@lps.ecnu.edu.cn; hbwu@phy.ecun.edu.cn

The energy transfer and heat flux play the most important role in thermodynamics, which have been well understood on macroscopic scales in the framework of classical thermodynamics. However, as the size of the system is reduced, thermal or quantum fluctuations becoming increasingly relevant, some striking phenomena and properties have been revealed recently[1–4]. To study heat transport in systems of nano/microscales and at single-atom levels has remained elusive, although it has received great interest for the fundamental physics in many fields[5–8] and the applications in various advanced microscopic devices[9–11].

In parallel, remarkable accomplishments have been achieved in cavity optomechanics, such as ground-state cooling of mechanical resonators[12–14], ultrasensitive motion detections[15–17], nonreciprocal control of photons and phonons[18–20], and so on. Optomechanical system owing to its high controllability has been proposed as a novel mechanism for nonlocal heat transfer and thermal management[21–26], however, it has not been experimentally demonstrated because of the technical challenges in precisely manipulating optomechanical arrays inside an optical cavity.

Here we study the transport of thermal energy between two spatially separated nanomechanical resonators mediated by the cavity light field. Compared to other mechanisms of heat transport, the optomechanical coupling can be coherently manipulated and it could be, in principle, infinitely long-range, which provides a new mechanism for heat transfer with high tunability. Importantly, in the strong coupling regime, we find that the heat flux flows back and forth coherently in nonequilibrium steady-state at short time scales, and the oscillation period is fully determined by the eigenmodes of such a composite system. This means that the coherence is dominated over the dissipations in such an inherently random process. In addition, we test the constraints on the precision of steady-state heat flux, i.e., the thermodynamic uncertainty relation[27–37], in such a thermal gradient driven far-off equilibrium system at various coupling strengths, including the strong coupling regime. Therefore, our observations provide new insights for nonequilibrium thermodynamics and nontrivial extension beyond the weak coupling, in contrast to most previous systems of stochastic thermodynamics that are overdamped[3,4].

## Results

**Experimental setup.** The multimode optomechanical system offers an exceptional degree of freedom to investigate phonon heat transport. The thermal energy is transported between two nanomechanical resonators mediated by a cavity field, as shown in Fig. 1a. The resonators are two SiN membranes with dimensions 1 mm × 1 mm × 50 nm. The membranes separated by 6 cm are placed inside a Fabry-Perot cavity[38,39]. The motions of membranes are monitored by two weak probe laser fields separately (not shown in Fig. 1a). We focus on the vibrational (1,1) modes, which are nearly degenerate with eigenfrequencies $\omega_{1,2} \sim 2\pi \times 400$ kHz. Piezos are used to precisely control the eigenfrequencies of each membrane[40]. The cavity is driven by a red-detuned laser field, which interacts with both membranes simultaneously due to the dynamical backaction. Therefore, two individual modes of membranes are effectively coupled mediated by the cavity field. The phonon heat transport is realized by contacting two membranes with independent thermal baths. Such a system can be equivalently viewed as two mutually coupled harmonic oscillators, as shown in Fig. 1b, with one in contact with a room temperature reservoir and the other with a high temperature reservoir (realized by exciting the membrane with a Gaussian white noise). The temperature gradient drives the system out of equilibrium with a mean heat flux from the hot reservoir to the cold reservoir under the effective optomechanical coupling. When the coupling between the membranes is zero by

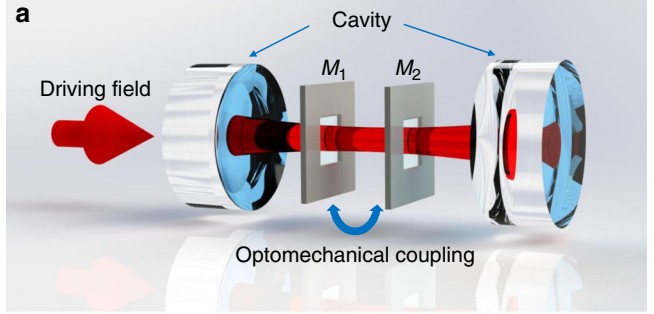

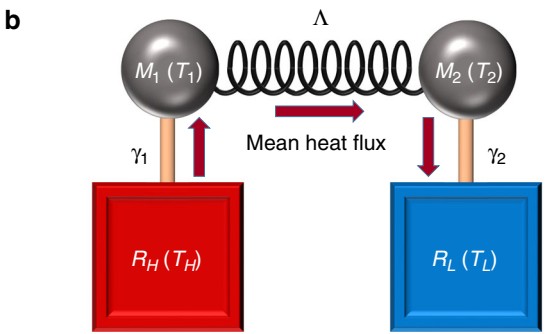

**Fig. 1 Heat transport between two optomechanically coupled nanomechanical resonators. a** Two nanomechanical SiN membranes ($M_1$ and $M_2$) are placed inside an optical cavity and coupled to a single cavity field via optomechanical backaction. **b** Equivalent model of two harmonic oscillators mutually coupled with a strength $\Lambda$ and each in contact with an independent thermal reservoir, i.e., $R_H$ and $R_L$ for high and low temperature reservoirs, respectively. $\gamma_1 = 2\pi \times 12$ Hz and $\gamma_2 = 2\pi \times 6$ Hz are the damping rates of the oscillators to their own reservoirs. $T_i$ ($i = 1, 2, H, L$) represents the temperature.

turning off the cavity field, the membranes have the equivalent temperatures as their own reservoirs.

**Effective Hamiltonian.** The interaction Hamiltonian of such a composite system of two membranes interacting with a common cavity field is $H_{int} = -\hbar \sum_{i=1,2} g_i \hat{a}^\dagger \hat{a} (\hat{b}_i^\dagger + \hat{b}_i)$, where $\hat{a}$ and $\hat{b}_i$ are the annihilation operators for cavity and mechanical modes, respectively. $g_i$ is the optomechanical coupling rate of each membrane. After eliminating the cavity mode, the system can be effectively described by the following Langevin equations (see Supplementary Note 1)

$$i\frac{\partial}{\partial t}\begin{pmatrix}\hat{b}_1 \\ \hat{b}_2\end{pmatrix} = \begin{pmatrix}\omega_1 - \frac{i\gamma_1}{2} + \Lambda & \Lambda \\ \Lambda & \omega_2 - \frac{i\gamma_2}{2} + \Lambda\end{pmatrix}\begin{pmatrix}\hat{b}_1 \\ \hat{b}_2\end{pmatrix} + \begin{pmatrix}\sqrt{\gamma_1}\hat{\eta}_1 \\ \sqrt{\gamma_2}\hat{\eta}_2\end{pmatrix}$$

(1)

Here $\hat{\eta}_i$ ($i = 1, 2$) is the thermal Langevin noise operator with $\langle \hat{\eta}_{1,2}^\dagger(t)\hat{\eta}_{1,2}(t')\rangle = \delta(t - t')k_B T_{H,L}/\hbar\omega_{1,2}$. The mechanical resonant frequencies are assumed to be equivalent, i.e., $\omega_1 = \omega_2 = \omega_0$, which leads to the maximum heat transfer efficiency, unless it is specifically described. $\Lambda = g^2 \chi_m$ is the effective coupling rate between membranes, where the membranes are assumed to have the same optomechanical coupling rate $g$, and $\chi_m$ is the effective mechanical susceptibility introduced by the intracavity field[41], which is proportional to the intracavity photon number. Generally, the effective coupling between membranes can be both dispersive and dissipative. Here, the laser frequency is detuned far-off the cavity resonance, and consequently the interacting of membranes is dominated by a conservative coupling.

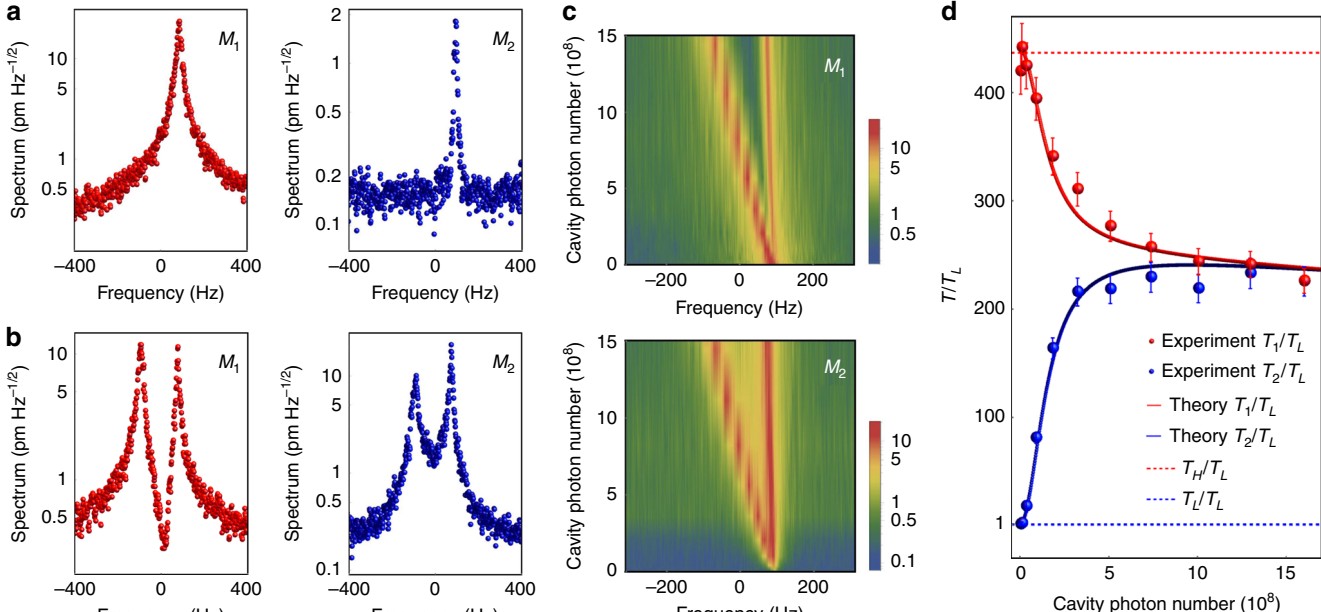

**Fig. 2 Normal-mode splitting and effective temperatures. a, b** Thermomechanical noise spectra of membranes in the weak and strong coupling regimes, respectively. $M_1$ and $M_2$ represent the membranes with high and low temperature reservoirs, respectively. The zero frequency represents the center of the normal-mode splitting, which is ~400 kHz. **c** Normal-mode splitting of the thermomechanical noise spectra as a function of cavity photon number. **d** Effective temperatures of membranes as a function of cavity photon number. Solid curves are the theoretical calculations and the dashed lines represent the reservoirs' temperatures. Each data point is an average over a measurement period of 50 s and the error bars are the standard deviations.

**Thermomechanical noise spectrum**. To demonstrate the thermal energy transportation, we first investigate the thermomechanical noise spectral properties. Figure 2a, b illuminate the noise spectra of membranes in the weak and strong coupling regimes, respectively. The critical point that separates the strong and weak coupling regimes is $\Lambda^2 - \frac{(\gamma_1 - \gamma_2)^2}{16} = 0$. Notably, the mechanical spectrum shows two peaks in the strong coupling regime, and the resonant frequencies are given by the eigenvalues of the effective Hamiltonian, i.e., $\omega_+ = \omega_0$ and $\omega_- = \omega_0 + 2\Lambda$, which is called the normal mode splitting. This splitting is caused by the strong phonon-phonon interaction, and it has a different mechanism with the previous observations where one cavity mode and one mechanical mode interact[42,43]. The frequency shifts of normal modes as a function of the intracavity photon number are plotted in Fig. 2c. Due to the effective spring constant is negative and the eigenfrequencies of membranes are also modified by the optomechanical coupling, the frequency of the breathing mode is fixed and the frequency of the center-of-mass mode reduces as the cavity photon number increases, as shown in Fig. 2c. By integrating the noise power spectra, one can obtain the effective temperatures of membranes. The root mean square vibrational amplitude of membrane $\sqrt{k_B T / m \omega_0^2}$ at room temperature is utilized to calibrate the effective temperature (see Supplementary Note 4). Please note that the effective temperature is defined by the motion of mechanical oscillator, instead of the motions of atoms which constitute the membrane. When the optomechanical coupling is turned on, the heat energy is transferred from the high temperature membrane to the low temperature membrane. The effective temperatures in steady states are shown in Fig. 2d. When the coupling is weak, the temperature difference between two membranes is large. The temperatures are close to their respective reservoirs' temperatures in equilibrium. As the cavity photon number increases, the temperatures of membranes tend to be equal, which implicates that the thermal energy has been redistributed in the two membranes under the optomechanical coupling.

**Instant heat flux**. To fully characterize the process of heat transfer, we investigate the heat flux in the nonequilibrium steady-state. The instant heat flux from membrane 1 to 2 is $j_\tau = -\lim_{t_0 \to \infty} \frac{2m}{\tau} (\omega_0 \Lambda + \Lambda^2) \int_{t_0}^{t_0 + \tau} u_1 \dot{u}_2 dt$, where $u_{1,2}$ is the membrane motion, and $\tau$ is the integration time[44,45]. Since the integration time of the measurements (typical on the order of 100 μs) is much smaller than the damping rates of membranes, then the heat flux directly obtained from the data of lock-in amplifier can be viewed as the instant heat flux. The time evolution of instant heat flux in the strong coupling regime is plotted in Fig. 3a. Remarkably, the instant heat flux oscillates back and forth between the membranes in the nonequilibrium steady-state with strong coupling (see the inset of Fig. 3a for clarity), which is not previously observed in the thermalization and in contrast with the observations in the macroscopic world. By performing Fourier transformation on the heat flux trace, the oscillation frequency as a function of cavity photon number is presented in Fig. 3b, which is coincident with the twice of the effective coupling strength between membranes. Such oscillations disappear as the coupling strength is smaller than the critical point. The inset of Fig. 3b illuminates such a transition near the critical point.

The probability density functions for various integration time are plotted in Fig. 3d. Although the instant heat flux traces exhibit completely different behaviors in the strong coupling regime, the probability density functions have exponential tails and positive mean values, which is consistent with the observations in overdamped systems[4]. Based on the probability density function, a symmetry function can be defined as the logarithm of the ratio of the probability finding a positive heat flux with respect to the corresponding negative one, i.e., $\ln[P(j)/P(-j)]$[40]. The symmetry function at different coupling strengths is shown in Fig. 3c, which has a linear dependence on the instant current value j. The green dots and brown squares are the experimental data corresponding to the cases when the cavity photon numbers are $5 \times 10^8$ and $10^9$, respectively. A larger optomechanical coupling strength leads to a

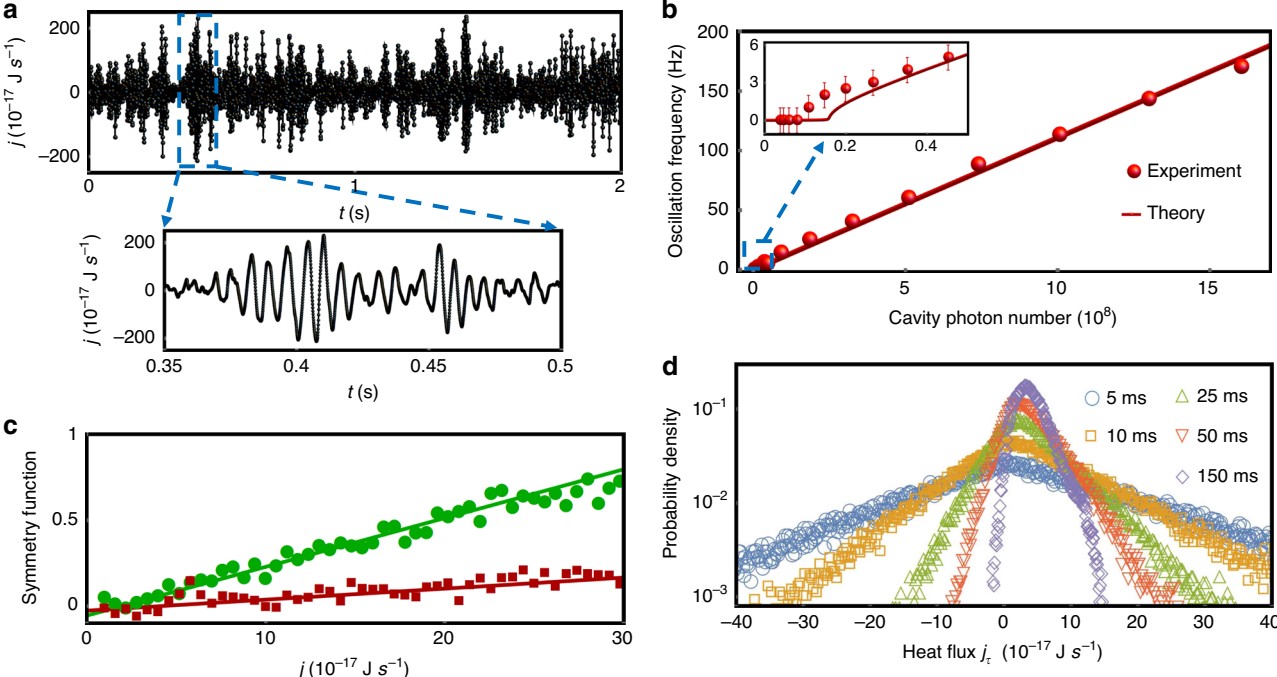

**Fig. 3 Instant heat flux. a** Time evolution of instant heat flux. The inset is the detailed oscillating instant heat flux for the purpose of clarity. **b** Coherent heat flux oscillation frequency as a function of cavity photon number. The inset is for the case of small cavity photon number, which exhibits a transition near the critical point. **c** Symmetry function at different coupling strengths. The lines are the fitting. **d** The corresponding probability density functions at different integration time.

smaller slope of the symmetry function, and indicates a more symmetric probability density function, which depends on the difference of the effective temperatures of membranes.

**Thermodynamics uncertainty relation**. According to the measurements of instant heat flux for the strong coupling (Fig. 3) and weak coupling cases (see Supplementary Note 5), one can find that the dynamical fluctuation is an essential quantity, which indicates that such an optomechanical system driven by a thermal bias provides an ideal platform for studying the theories in nonequilibrium thermodynamics, the efficiency of stochastic heat engine, and the connection between information and thermodynamics[3,4]. The instant heat flux in the strong coupling regime could violate the second law of thermodynamics in the small time scales. Then we intend to investigate whether the fundamental constraints of stochastic thermodynamics hold in this distinct regime. The thermodynamics uncertainty relation (TUR) is a recently developed thermodynamic inequality, which offers a precise bound on the heat flux fluctuations in terms of the entropy production. Although the TUR has been well accepted and the most interesting applications take a step beyond its validation, there is rare proof in the experiment[30,37].

The statement of the TUR in the nonequilibrium steady-state limit under a constant thermal bias reads[27–37]

$$\frac{\text{Var}(J_\tau)}{J_\tau^2} \geq \frac{2k_B}{\sum_\tau} \quad (2)$$

where the average steady-state entropy production is $\sum_\tau = \langle J_\tau \rangle \left( \frac{1}{T_L} - \frac{1}{T_H} \right)$ and the integrated heat current is $J_\tau = -\lim_{t_0 \to \infty} 2m(\omega_0\Lambda + \Lambda^2) \int_{t_0}^{t_0+\tau} u_1 \dot{u}_2 dt$. The trajectories of integrated heat current as a function of time at different coupling strengths are plotted in Fig. 4a. In the case of $T_L \ll T_H$, Eq. (2) simplifies to $\text{Var}(J_\tau)/(\langle J_\tau \rangle k_B T_L) \geq 2$. Figure 4b exhibits the TUR quantity $\text{Var}(J_\tau)/$

$(\langle J_\tau \rangle k_B T_L)$ as a function of $\tau$ at various coupling strengths. As one can see in Fig. 4b, the TUR quantity is above the TUR limit (a value of 2) at an arbitrary time, which approaches to a constant as $\tau$ is relatively large. However, the TUR decreases as $\tau$ goes to zero, owing to the possibility that the inertia suppress the fluctuations when the time is smaller than the relaxation time of membranes. A characteristic behavior for the strong coupling case (see the inset of Fig. 4b for clarity) is that the TUR quantity shows oscillation and the period is coincident with the oscillation period of instant heat flux.

**Mean heat flux**. Lastly, we demonstrate that the mean heat flux can be flexibly manipulated by tuning the laser power and other parameters, in comparison with many other systems where the thermal conductivity is fixed and the heat flux is difficult to tune. The value of mean heat flux is obtained by averaging the instant heat flux trace for a long time (dozens of seconds in contrast to hundreds of μs for the instant heat flux), which can also be extracted from the effective temperatures of membranes in the nonequilibrium steady-state (see Supplementary Note 2). The mean heat flux is zero when the laser field is off. As the intracavity photon number increases, the mean heat flux grows up and tends to be saturated, as shown in Fig. 5a. The maximum mean heat flux is limited by $\gamma_1\gamma_2 k_B(T_H - T_L)/(\gamma_1 + \gamma_2)$. The mean heat flux is linearly related to the temperature difference of two reservoirs, i.e. $T_H - T_L$, which is depicted in Fig. 5b. Previously, we have studied the situations of resonant heat transfer. Figure 5c presents the dependence on the frequency detuning of membranes ($\Delta = (\omega_1 - \omega_2)/2\pi$). The resonance behavior can be clearly seen in Fig. 5c. Interestingly, the linewidth of the resonance peak is broadened compared to the natural linewidth of mechanical resonators, which can be analogous to the power broadening in atomic physics and can be utilized to relax the condition of mechanical frequency match in the strong coupling regime.

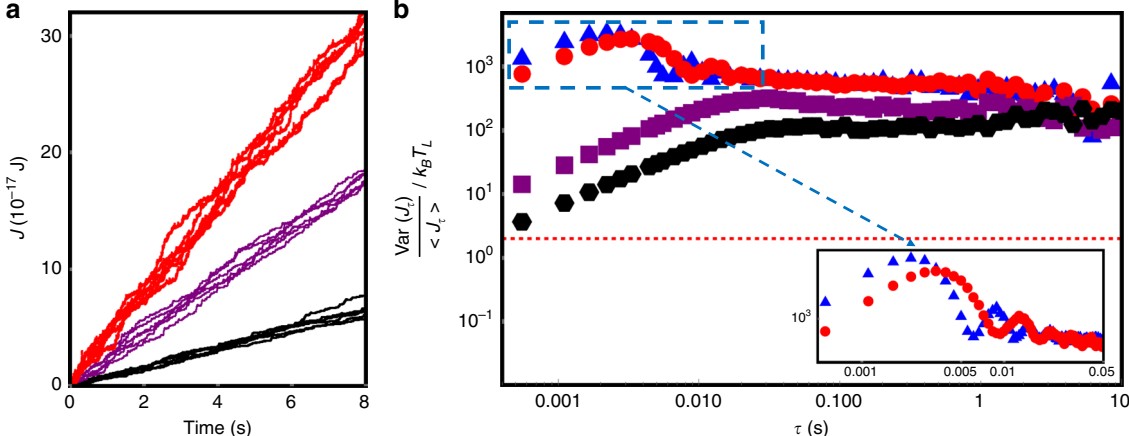

**Fig. 4 Thermodynamics uncertainty relation (TUR). a** Trajectories of integrated heat current as a function of time at different coupling strengths. The black, purple, and red curves are for the coupling strengths $\Lambda/2\pi = 6$, 8, and 58 Hz, respectively. **b** The TUR quantity, $\mathrm{Var}(J_\tau)/(\langle J_\tau \rangle k_B T_L)$, as a function of integration time $\tau$ for various coupling strengths. The black hexagons, purple squares, and red dots have the corresponding coupling strengths as in (**a**). The blue triangles are for the coupling strength ~85 Hz. The red dashed line represents the TUR limit.

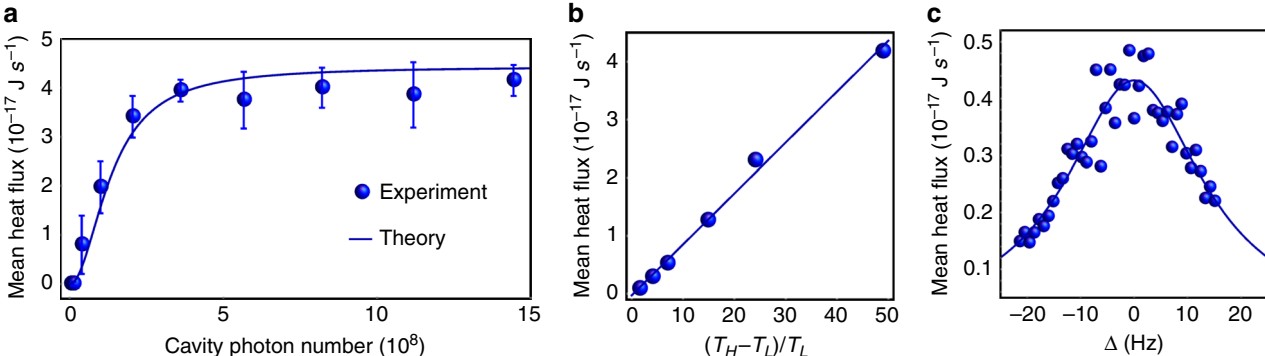

**Fig. 5 Mean heat flux. a** Mean heat flux as a function of cavity photon number. **b** Mean heat flux as a function of temperature difference. **c** Mean heat flux as a function of frequency detuning of membranes.

## Discussion

The systems of optomechanical arrays offer many unique possibilities to study nonequilibrium dynamics, as they allow for a full real-time control of almost all relevant parameters. We have demonstrated the phonon heat transportation between two spatially separated nanomechanical membranes with cavity-enhanced long-rang and flexible optomechanical coupling. The spontaneous generation of coherent energy oscillation during the process of heat transfer with strong coupling is experimentally observed. The universal bound on the precision of nonequilibrium steady-state heat flux is verified. In contrast to most previous stochastic thermodynamic systems, which are over-damped, our observations might open an unexplored region. Although one eigenmode of each membrane is utilized in this work, the multimode nature of the mechanical resonator implies the possibility of using this system as a multi-channel phonon-photon interface[46]. Such a system provides a fertile ground for studying stochastic and quantum thermodynamics, and is a candidate for the photon controlled phononic devices and hybrid quantum networks.

## Data availability

The data that supports the results within this paper are available from the corresponding authors upon reasonable request.

## Code availability

The custom code used to obtain the results within this paper are available from the corresponding authors upon reasonable request.

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

## Acknowledgements

We thank Jordan M. Horowitz for helpful discussions. This research was supported by the National Key R&D Program of China (No. 2017YFA0304201), NSFC (No. 11925401, No. 11734008, No. 11974115, and No. 11704126, No. 11621404), the Shanghai Committee of Science and Technology (No. 17JC1400500), the Program for Professor of Special Appointment (Eastern Scholar) at Shanghai Institutions of Higher Learning.

## Author contributions

C.Y., X.W., J.S., and H.W. carried out the experiment, analyzed the data, and developed the theory. J.S. and H.W. wrote the paper with input from all authors.

## Competing interests

The authors declare no competing interests.
