## [Peer Review File · Nature Communications]

Reviewers' Comments:

Reviewer #1:

Remarks to the Author:

Yang and coauthors show the Rabi-like oscillation of nonlocal heat transfer in a system having two mechanical modes mediated by a single optical mode, which is called a multimode optomechanical system. The two mechanical modes reside at different temperatures, and the mode at a higher temperature starts to transfer its energy to the other when turning on the optomechanical coupling. Interestingly, when the system reaches into the strong coupling regime, the heat flux between the modes begin to oscillate while the net heat flux is flowing from the higher to lower temperature mode. The authors clearly demonstrate the experimental data and supportive theoretical analysis.

Overall, I believe the reported work has the standards that one may expect for a publication in Nature Communications. The manuscript is well organized and clearly written with a detailed description of all the aspects of a new experiment in optomechanics, which can stimulate further works. Still, before I fully recommend the manuscript to Nature Communications, I would like to suggest the authors answer my questions and comments.

1. Figure 2 d shows that the effective temperature of M1 is two orders of magnitude larger than that of M2, which is insanely large. Both of the membranes are initially at room temperature, and the authors make M1 heated by driving white noise via PTZ, as illustrated in the Supplement S5. It is hard to believe that the white noise from the piezoelectric block heats a small membrane up to more than 100 times, even the authors focus on effective temperatures. I understand effective temperatures are calculated from integrating mechanical mode PSDs over the area. However, it is not unclear to me how to calibrate measured spectra at the spectrum analyzer for obtaining the mechanical mode PSDs, i.e., $S_{\{b^{\dagger}b\}}(\omega)$, thereby how to extract each effective temperatures at this point.

2. In Fig. 3a, the instant heat flux is represented as a function of the probability density. Intuitively, it should look skewed, or at least the medium value must be at off-axis, but it seems quite symmetric at zero instant heat flux. Could you elaborate on what happened in the instant heat flux thoroughly? For instance, how skewed the curve is and or what is the center value?

Minor comments:

In the caption of Figure 2 a.b, the authors mention, "The zero frequency represents the intrinsic frequency of membranes, which is ~ 400 kHz." As shown in Fig 2.a, however, the center frequencies of M1 and M2 spectrums are located at not zero where they are supposed to be. Rather the zero frequency looks like the center frequency of two normal modes after showing normal-mode splitting.

Reviewer #2:

Remarks to the Author:

This article presents an experimental study of energy transport between optomechanical resonators mediated by a cavity mode. The authors report several observations, including an oscillation of instantaneous heat flux between the oscillators and a bound on the thermodynamic uncertainty relation.

The authors have realized a novel experimental platform that can be used to study thermally driven coupled mechanical oscillators and associated phenomena in non-equilibrium thermodynamics. However, I hesitate to recommend it for publication in Nature Communications in its present form for the following reason:

-The primary results of the paper follow from what is expected from the dynamics of two coupled oscillators and thus are not particularly surprising. The instantaneous heat flux oscillation is

expected given that the displacements of the membranes (driven by broadband thermal forcing) oscillate in time. The claims that the result is a "previously undiscovered phenomenon for heat transfer" and that the platform could enable tests of the fundamental laws of physics are a stretch, particularly the former as energy transfer between coupled oscillators is a textbook topic. On the other hand, the experimental evidence supporting the thermodynamic uncertainty relation is an interesting result that supports the theory of non equilibrium thermodynamics. I believe the paper could be strengthened by putting additional focus on this result and further elaborating on what other tests of non-equilibrium thermodynamics could be performed using their setup. Tests of e.g. "connection between fluctuations and response" are less compelling since they have been thoroughly investigated theoretically and experimentally. It is possible, with these changes and a rephrasing of the claims as described above, that the resulting manuscript could be suitable for publication in Nat Comms.

Reviewer #3:

Remarks to the Author:

The authors in this paper have studied the dynamics of two membranes that are mechanically coupled through a laser field, and are each attached to a thermostat.

Through a variation in the number of photons in the cavity, they can change the strength of the coupling between the two membranes, thereby controlling the flow of heat between the two.

According to their equations of motion, the dynamics is linear, meaning the equations of motion are linear in membrane displacements. Furthermore, the noise from the Langevin thermostats satisfies the fluctuation-dissipation (FD) theorem. As a result, if one uses the Green's function method to solve the equations of motion, one finds in the long time limit: $x(\omega) = G(\omega) \eta(\omega)$. Given the FD, on the other hand, $\langle \eta(\omega_1) \eta(\omega_2) \rangle = \delta(\omega_1 + \omega_2)$. As a result, the heat current being quadratic in $x(\omega)$ cannot have a Fourier (AC) component and can only be static. Yet the measurements show an oscillatory behavior. Therefore the theory used in this paper does not seem to be consistent with the experimental observations! I do not understand where the deterministic equations in section 9 of the supplementary material came from. What happened to the noise terms present in S12?

This brings me to the question of the measurement of heat. What is plotted as heat is from the measured displacements. It maybe called the energy transferred or exchanged between the two oscillators, which is really a mechanical energy. Is that the same as "thermodynamic heat"? In Figure S4, in fact, we see temperatures on the order of 100,000K. These are not thermal energies but simply kinetic energies. One could drive two coupled macroscopic oscillators and have energy go back and forth between them, but I would not call it heat. If this is really heat, then one should be able to perform cooling, i.e. pumping heat from the cold membrane to the hot one, and bring its temperature even lower. Can this be achieved in this setup?

It is not clear how the temperatures of the membranes are measured? The effective temperature obtained from the square of displacements is just a kinetic energy and in fact is as large as 130000K.

Other comments:

-The power spectrum S introduced in the supplementary material was never defined.

-the factor k_B should be removed from the definition of the entropy generation rate in the line following eq (2). The equation 2 does not seem to have the correct units. A timescale is missing. 4 lines lower, where the ratio of variance to current is expressed, again, a timescale seems to be missing. J is an energy per unit time.

Response to the referee reports of manuscript NCOMMS-20-11938-T by Yang et al:

We thank the referees for their careful reading of the manuscript and constructive suggestions. The Reviewer 1 believes the reported work has the standards that one may expect for a publication in Nature Communications and can stimulate further works. The Reviewer 2 thinks that the manuscript could be suitable for publication in Nature Communications with some changes and a rephrasing of the claims. The main comments of the Reviewer 3 is regarding to the consistency of the experimental observations and theoretical analysis, as well as the concept of “thermodynamic heat”. We feel that the concerns from the Reviewer 3 are due to some lack of clarity in our presentation.

We have fully taken into account the referees’ suggestions to improve the manuscript. In the following, we will respond to all comments from the referees and indicate changes made in the revised manuscript.

Response to the comments of the Reviewer 1:

(1) Comments: “Interestingly, when the system reaches into the strong coupling regime, the heat flux between the modes begin to oscillate while the net heat flux is flowing from the higher to lower temperature mode. The authors clearly demonstrate the experimental data and supportive theoretical analysis. Overall, I believe the reported work has the standards that one may expect for a publication in Nature Communications. The manuscript is well organized and clearly written with a detailed description of all the aspects of a new experiment in optomechanics, which can stimulate further works.”

Response: We would like to thank the Reviewer 1 for his/her positive evaluation, finding the manuscript “has the standards that one may expect for a publication in Nature Communications.”

(2) Comments: “Figure 2 d shows that the effective temperature of M1 is two orders of magnitude larger than that of M2, which is insanely large. Both of the membranes are initially at room temperature, and the authors make M1 heated by driving white noise via PTZ, as illustrated in the Supplement S5. It is hard to believe that the white noise from the piezoelectric block heats

a small membrane up to more than 100 times, even the authors focus on effective temperatures. I understand effective temperatures are calculated from integrating mechanical mode PSDs over the area. However, it is not unclear to me how to calibrate measured spectra at the spectrum analyzer for obtaining the mechanical mode PSDs, i.e., $S_{b^{\dagger}b}(\omega)$, thereby how to extract each effective temperatures at this point.”

Response: We agree with the referee that it looks difficult to achieve the effective temperature of M1 two orders of magnitude larger than that of M2 just by driving white noise via PTZ. However, it is possible in the field of optomechanics. This is because the effective temperature is defined by the root-mean-square (rms) displacement $x_{rms} = \sqrt{\langle x^2 \rangle} = \sqrt{k_B T^{eff} / m\omega^2}$. Although the effective temperature is two orders of magnitude larger than the room temperature, the rms displacement is just changed one order of magnitude. Before presenting the detailed calibration of an effective temperature of 130000K in our case, we first use the cantilever as an example [EPL 89, 60003 (2010)] to demonstrate the possibility to make the effective temperature of a micro/nano-mechanical resonator more than 100 times. According to the parameters of cantilever given in the above paper, the rms displacement is ~ 0.1 nm at room temperature. In order to obtain an effective temperature of 130000K, the rms displacement of the cantilever should be driven up to a few nanometers. As far as we know, the rms displacement of a cantilever can be driven at least two or three orders of magnitude larger than the one at room temperature [Appl. Phys. Lett. 78, 1637 (2001)]. Similarly, the rms displacement of nanomechanical membrane used in our experiment at room temperature and high temperature (130000K) are 4.5 pm and 94 pm, respectively, due to the relatively large eigenfrequency, ω . In our current experimental setup, 100 pm rms displacement is a value which can be achieved by driving white noise via PZT. Actually, the displacement of the membrane can be driven even larger [AIP Advances 8, 015209 (2018)]. In the following, we will give more details regarding the method to extract each effective temperature.

The effective temperature can be obtained both from the measurements by the spectrum analyzer and the lock-in amplifier. We calibrate the effective temperatures with the data of spectrum analyzer by integrating PSD over the area (by fitting the experimental data first and

finding the ratio of the integrated areas by comparing with the room temperature data). We also use the data of lock-in amplifier to calibrate the effective temperatures. The results obtain from these two methods are consistent with each other. For example, by using the spectrum analyzer, the noise power spectra at room temperature and high temperature (with additional white noise driving) are shown in Figs. R1(a) and R1(b), respectively. The integrated areas for Figs. R1(a) and R1(b) are obtained approximately as 67 and 3×10^4 in arbitrary units but in the same procedure, respectively. Figure R1(c) presents the table of the integrated areas for different integration ranges. We know the room temperature is $\sim 300\text{K}$. Therefore, we can obtain that the effective temperature of the membrane in Fig. R1(b) is $\sim 130000\text{K}$.

Fig. R1. (a) and (b) present the power spectral density for the cases of room temperature and high temperature, respectively. The blue and red dots are the experimental data. The black curves are the fitting. The grey shadows are the areas for the integration. (c) is the table for various integration ranges, in order to demonstrate that the integration from -400 to +400 Hz is reasonable to estimate the effective temperature.

In the following, we show how to use lock-in amplifier data to calibrate the effective temperature, because this way is more convenient and straightforward in the experiment, and we have used this way for most figures in the manuscript. From the measurements of lock-in amplifier, we can directly obtain two quadratures, i.e. X and Y (please see Fig. S4(a) in the supplementary). According to the equipartition theorem, i.e., $k_B T^{eff} = m\omega^2 \langle x^2 \rangle = m\omega^2 \langle X^2 + Y^2 \rangle / 2$, we can directly obtain the effective temperature with $T_{High}^{eff} = T_{Room}^{eff} \langle X_{High}^2 + Y_{High}^2 \rangle / \langle X_{Room}^2 + Y_{Room}^2 \rangle$. Here T_{High}^{eff} and T_{Room}^{eff} are the effective temperatures for the cases of high temperature and room temperature, respectively.

In conclusion, we have responded the referee's comments by explaining the procedures of calibration with both measurements from the spectrum analyzer and lock-in amplifier. The effective temperature can be obtained from the integrated area of PSD, or from the variance of the displacement. The effective temperature with two orders of magnitude larger than the room temperature looks insane, while the rms displacement at the high temperature is approximately one order of magnitude larger than the one at the room temperature, which can be achieved in our current experimental setup. More details are added in the supplementary to make the calibration procedure more clear.

(3) **Comments:** "In Fig. 3a, the instant heat flux is represented as a function of the probability density. Intuitively, it should look skewed, or at least the medium value must be at off-axis, but it seems quite symmetric at zero instant heat flux. Could you elaborate on what happened in the instant heat flux thoroughly? For instance, how skewed the curve is and or what is the center value?"

Response: It is true that the probability distribution function in Fig. 3a should be skewed and the medium value should be off-axis. In the following, we will respond this comment in two aspects (i) the medium value, and (ii) the skewness.

(i) We find that the medium value of the probability distribution depends on the integration time. Figures R2 show the probability density function at different integration time. This result can be qualitatively understood that the medium value of the distribution is closer to the mean value as the integration time is longer. Our results are consistent with previous observations from other

groups, e.g. EPL 89, 60003 (2010). The probability density in Fig. 3 in the manuscript is replaced by Fig. R2.

Fig. R2. Probability density function at different integration time.

Fig. R3. Symmetry function at different coupling strength. The green stars and brown triangles are the experimental data corresponding to the cases when the cavity photon numbers are 5×10^8 and 10^9 , respectively. The green and brown lines are the fitting.

(ii) To analyze the skewness of the probability distribution function, we adopt the definition of symmetry function in a previous work by A. Berut et al [Phys. Rev. Lett. 116, 068301 (2016)]. The symmetry function or the skewness at different coupling strength is plotted in Fig. R3. The probability density function becomes more symmetric as the coupling strength increases (the slope of the symmetry function is smaller when the probability distribution function is more symmetric). Considering the effective temperature of membranes as a function of the coupling strength shown in Fig. 2(d) in the manuscript, one can find that the skewness of the probability

distribution function depends on the effective temperature difference of the membranes. Our results are consistent with the ones presented in Phys. Rev. Lett. 116, 068301 (2016). In that work, the skewness depends on the effective temperature difference of the baths. Since the system in that work is overdamped, then the effective temperature of the oscillator is the same as the bath. Therefore, our results reveal that the skewness is actually depending on the temperature difference of the oscillators.

In conclusion, we have updated Fig. 3 and added more discussions in the manuscript to elaborate on what has happened in the instant heat flux thoroughly.

(4) **Comments:** “In the caption of Figure 2 a.b, the authors mention, “The zero frequency represents the intrinsic frequency of membranes, which is ~ 400 kHz.” As shown in Fig 2.a, however, the center frequencies of M1 and M2 spectrums are located at not zero where they are supposed to be. Rather the zero frequency looks like the center frequency of two normal modes after showing normal-mode splitting.”

Response: We thanks the referee for pointing out this to us. In the revised manuscript, the sentence has been modified as “the zero frequency represents the center of the normal-mode splitting, which is ~ 400 kHz.”

Response to the comments of the Reviewer 2:

(1) **Comments:** “The authors have realized a novel experimental platform that can be used to study thermally driven coupled mechanical oscillators and associated phenomena in non-equilibrium thermodynamics.”

Response: We would like to thank the Reviewer 2 for this positive comment.

(2) **Comments:** “The primary results of the paper follow from what is expected from the dynamics of two coupled oscillators and thus are not particularly surprising. The instantaneous heat flux oscillation is expected given that the displacements of the membranes (driven by broadband thermal forcing) oscillate in time. The claims that the result is a “previously undiscovered

phenomenon for heat transfer” and that the platform could enable tests of the fundamental laws of physics are a stretch, particularly the former as energy transfer between coupled oscillators is a textbook topic.”

Response: In response to the comments, we have modified the claim of “previously undiscovered phenomenon for heat transfer” into “previously unobserved phenomenon for heat transfer with mechanical oscillators” in the revised manuscript. We agree with the referee that energy transfer between coupled oscillators is a textbook topic, and the studies of nonequilibrium thermodynamics are more interesting than the oscillatory heat current itself. Although the physics behind the oscillatory heat current can be understood straightforwardly, we did not find similar experimental results for the mechanical oscillators by carefully going through the literatures. It shares the same model of the Rabi oscillation, in which the membrane can be excited coherently instead by a thermal force. We think that the Rabi oscillation is not surprising, while the oscillation in the heat current might be interesting, because the oscillation is not wiped out by the random process and it starts spontaneously. More importantly, it is the signature of strong coupling for phonon heat transfer.

(3) **Comments:** “On the other hand, the experimental evidence supporting the thermodynamic uncertainty relation is an interesting result that supports the theory of non-equilibrium thermodynamics. I believe the paper could be strengthened by putting additional focus on this result and further elaborating on what other tests of non-equilibrium thermodynamics could be performed using their setup. Tests of e.g. “connection between fluctuations and response” are less compelling since they have been thoroughly investigated theoretically and experimentally. It is possible, with these changes and a rephrasing of the claims as described above, that the resulting manuscript could be suitable for publication in Nat Comms.”

Response: We thank the referee’s positive comments and useful suggestions. Taken the referee’s advice, we have put addition information and discussions on the thermodynamic uncertainty relation and the theory of non-equilibrium thermodynamics. For example, figure 3 in the manuscript is modified and additional discussions are added to address the fluctuating properties of the instant heat current. Moreover, figure 4 in the manuscript has been modified by including

more experimental results, as shown in Fig. R4. Please see more details in the revised manuscript. Tests of “connection between fluctuations and response” have been removed. The last sentence of the abstract in the revised manuscript has been replaced by “Our results reveal a previously unobserved phenomenon for heat transfer with mechanical oscillators, and provide a playground for testing fundamental theories in non-equilibrium thermodynamics.” There are several advantages of using such an optomechanical system to study non-equilibrium thermodynamics, e.g. flexible controllability, precise measurements, and the ability to operate deep in quantum regime. The following discussion has been added in the revised manuscript to elaborate on what other tests of non-equilibrium thermodynamics could be performed using this setup: “such an optomechanical system driven by a thermal bias provides an ideal platform for studying the nanoscale stochastic heat engine, the connection between information and thermodynamics, and thermalisation in nonequilibrium thermodynamics, ”.

In conclusion, we have putting additional focus on non-equilibrium thermodynamics by adding more experimental results and discussions. In addition, some claims have been rephrased according to referee’s suggestion.

Fig. R4. Thermodynamic uncertainty relation (TUR). a, Trajectories of integrated heat current as a function of time at different coupling strengths. b, The TUR quantity as a function of integration time τ for various coupling strengths. The red dashed line represents the TUR limit.

Response to the comments of the Review 3:

(1) **Comments:** “According to their equations of motion, the dynamics is linear, meaning the equations of motion are linear in membrane displacements. Furthermore, the noise from the Langevin thermostats satisfies the fluctuation-dissipation (FD) theorem. As a result, if one uses the Green's function method to solve the equations of motion, one finds in the long time limit: $x(w)=G(w) \eta(w)$. Given the FD, on the other hand, $\langle \eta(w_1) \eta(w_2) \rangle = \delta(w_1+w_2)$. As a result, the heat current being quadratic in $x(w)$ cannot have a Fourier (AC) component and can only be static. Yet the measurements show an oscillatory behavior. Therefore the theory used in this paper does not seem to be consistent with the experimental observations!”

Response: The measurement of the instant heat current shows an oscillatory behavior, while the referee calculates the mean heat current, i.e. the average of the instant heat current. This is the origin of the discrepancy. In other words, the mean heat current $\langle x_1(t)dx_2(t)/dt \rangle$ is static, however, the instant heat current $x_1(t)dx_2(t)/dt$ can fluctuate. In the following, we will give more details to support our point.

(i) We agree with the referee that the mean heat current cannot have a Fourier (AC) component and can only be static. This result has been shown in Fig. 5 in the manuscript. It is widely known that the statistical averages of the thermal force satisfy $\langle f(t) \rangle = 0$, $\langle f(t)f(t') \rangle = C\delta(t-t')$ (please see eq. (2.8) in J. Phys.: Condens. Matter 27, 214005 (2015) by Prof. Barton). While for each individual time, $f(t) \neq 0$. Therefore, the heat current can have a different value at each individual time. We would like to emphasize that, although the formula of the mean heat current (please see eq. 5.1 in Ref. [44]) and the formula of the instant heat current used in the manuscript look similar, they have different timescales. In the experiment, we choose the timescale of the instant heat current $\sim 100 \mu\text{s}$, while the timescale of the mean heat current is \sim second.

(ii) According to the recent progresses in the field of stochastic thermodynamics, it has been known that the instant heat current will fluctuate in time, which satisfies certain probability distribution functions, instead of being static. The fluctuating current can have an AC component. The contribution of our work is building up a new system which can operate in the strong coupling regime, therefore, there is a specific frequency component instead of broadband in contrast to

the previous experimental works. Some previous experimental works are, for example, the heat flux and entropy produced by thermal fluctuations have been investigated in electrical circuits [Phys. Rev. Lett. 110, 180601 (2013)], the stationary and transient fluctuation theorems for effective heat fluxes have been studied in hydrodynamically coupled particles in optical traps [Phys. Rev. Lett. 116, 068301 (2016)], and the distribution of entropy production have been studied in a single-electron box [Nature Physics 9, 644 (2013)].

In conclusion, we feel that the referee's argument is due to the lack of clarity of our presentations in the manuscript. We deal with the instant heat current instead of mean heat current. The observed mean heat current (Fig. 5) is consistent with the referee's comments. In the revised manuscript, we have added the difference between the instant and mean heat current, and their timescales to clarify this point.

(2) **Comments:** "I do not understand where the deterministic equations in section 9 of the supplementary material came from. What happened to the noise terms present in S12?"

Response: The equations in section 9 of the supplementary are for the purpose of analyzing experimental data directly from the lock-in amplifier. By using the data of lock-in amplifier, the mechanical displacement of each oscillator can be decomposed into $x_i(t) = X_i(t)\cos\omega_r t - Y_i(t)\sin\omega_r t$, where ω_r is the reference frequency of the lock-in amplifier, and $X_i(t)$ and $Y_i(t)$ are the quadrature components of the lock-in amplifier (please see the section 8 of the supplementary). ω_r , $X_i(t)$, and $Y_i(t)$ are the values which can be direction obtained from the experiment. The noise terms present in S12 are included in $X_i(t)$ and $Y_i(t)$. In conclusion, S12 is the equation derived from the Hamiltonian of the system, which is for the theoretical perspective. While the section 9 is the analysis based on the experimental measurements, which is for the experimental perspective. We have added additional information in section 9 of the supplementary to make this point more clear.

(3) **Comments:** "This brings me to the question of the measurement of heat. What is plotted as heat is from the measured displacements. It maybe called the energy transferred or exchanged between the two oscillators, which is really a mechanical energy. Is that the same as "thermodynamic heat"?? In Figure S4, in fact, we see temperatures on the order of 100,000K.

These are not thermal energies but simply kinetic energies. One could drive two coupled macroscopic oscillators and have energy go back and forth between them, but I would not call it heat. “

Response: The heat flux measured in the manuscript is defined as $\langle \dot{x}^2 \rangle / dt$. Such a definition has been used in several works in the field of stochastic thermodynamics. For example, Eq. 6 in Phys. Rev. Lett. 116, 068301 (2016) is given by analogy with the case of a single trapped Brownian particle; Eq. (5.1) in J. Phys.: Condens. Matter 27, 214005 (2015) is identified as the work done on one oscillator by the force exerted on it by the other oscillator. Therefore, it is the same as the thermodynamic heat and these are thermal energies. To further support our argument, we would like to present some experimental results from other groups. For example, Prof. Ciliberto's group achieves the effective temperature by sending a Gaussian white noise to the acousto-optic deflector in order to move the position of the optical trap randomly [Europhys. Lett. 107, 60004 (2014), Phys. Rev. Lett. 116, 068301 (2016)]. Therefore, in their works, the heat is from the measured displacement of the silica bead, and the effective temperature of a few thousand kelvin has been obtained in their works. The same method has also been used in several other groups, please see, e.g. Phys. Rev. Lett. 96, 070603 (2006), Nature Nanotechnology 9, 358 (2014), Nature Physics 11, 971 (2015), Phys. Rev. Lett. 114, 120601 (2015). In these works, the thermodynamic quantities are obtained from the trajectories of the mechanical oscillators, i.e. the displacements. Therefore, the definition of thermodynamic heat in the manuscript is consistent with previous works from other groups. We follow the standard way to calibrate the effective temperature, please see the response of the second comment of Reviewer 1 and the fifth comment of Reviewer 3 for more details. The energy transferred between two coupled oscillators has stochastic properties, which is the same as the thermodynamic heat used in several other works. In conclusion, we understand that it may be an open question to define heat and temperature in the stochastic thermodynamics and quantum thermodynamics. In this manuscript, we follow the well-accepted definitions of heat in the field of stochastic thermodynamics.

(4) **Comments:** “If this is really heat, then one should be able to perform cooling, i.e. pumping heat from the cold membrane to the hot one, and bring its temperature even lower. Can this be achieved in this setup?”

Response: Yes, it is heat, and it is able to be cooled in the system. Actually, recently we have done another experiment, in which the heat can be transfer from the hot one to the cold one, and the work is performed simultaneously, i.e. the so-called heat engine. As for pumping heat from the cold membrane to the hot one, this is an interesting question, because it is related to the heat pump or refrigerator. In order to pump heat from the cold membrane to the hot one, the Hamiltonian of the system should be engineered, for example, it can be achieved by additional driving, time-dependent system parameters, or modified heat baths [Phys. Rev. Lett. 118, 223602 (2017)]. In conclusion, the cooling by pumping heat from the cold one to the hot one mentioned by the referee can be achieved in this setup, while it requires further development.

(5) **Comments:** “It is not clear how the temperatures of the membranes are measured? The effective temperature obtained from the square of displacements is just a kinetic energy and in fact is as large as 130000K.”

Response: Either the membrane is driven by the thermal noise or the electric Gaussian white noise, the effective temperature can be obtained from the noise power spectrum. The noise power spectral density is defined as the Fourier transform of the autocorrelation function of the displacement, i.e. $S(\omega) = \int_{-\infty}^{+\infty} \langle x(t)x(0) \rangle e^{i\omega t} dt$. In the experiment, a typical noise power spectrum of the membrane is shown in Fig. R5(a). The area under the noise power spectrum yields the variance of the mechanical displacement, i.e. $\int_{-\infty}^{+\infty} S(\omega) d\omega / 2\pi = \langle x^2 \rangle$. According to the equipartition theorem, one can obtain $\langle x^2 \rangle = k_B T / m\omega_m^2$. Therefore, the effective temperature is obtained from the variance of displacements, rather than the square of displacements. Please note the difference of the situation shown in Fig. R5(b), in which the membrane is driven by a sine wave at its eigenfrequency instead of thermal noise. The noise power spectrum in Fig. R5(b) shows a delta-like function. Although it has a similar value of displacement as in Fig. R5(a), it is a coherent state rather than a thermal state, and we cannot define a temperature on it.

Fig. R5. Noise power spectra for a thermal state (a) and a coherent state (b), respectively.

(6) **Comments:** “The power spectrum S introduced in the supplementary material was never defined.”

Response: We thank the referee for this comment. We have defined the noise power spectrum in the revised manuscript. The definition of power spectrum is $S_{\hat{b}^\dagger \hat{b}}(\omega) = \int_{-\infty}^{+\infty} \langle \hat{b}^\dagger(\omega) \hat{b}(\omega') \rangle d\omega'$.

(7) **Comments:** “the factor k_B should be removed from the definition of the entropy generation rate in the line following eq (2). The equation 2 does not seem to have the correct units. A timescale is missing. 4 lines lower, where the ratio of variance to current is expressed, again, a timescale seems to be missing. J is an energy per unit time.”

Response: We thank the referee for pointing out this to us. A timescale is indeed missing in the formula. The definition of the average steady state entropy production rate should be

$\langle \sigma \rangle = k_B \Delta \beta \langle j_\tau \rangle = \left(\frac{1}{T_L} - \frac{1}{T_H} \right) \langle j_\tau \rangle$. Combined with the comments from the referee 2, we have

basically rewritten this part. We have defined the integrated heat current as

$J_\tau = - \lim_{t_0 \rightarrow \infty} 2m(\omega_0 \Lambda + \Lambda^2) \int_{t_0}^{t_0 + \tau} u_1 \dot{u}_2 dt$. Therefore, in the revised manuscript, j is an energy per unit time and J is an energy. We also follow the thermodynamics uncertainty relation at finite-time

[Phys. Rev. E 96, 012101 (2017); Nature Physics 16, 15 (2020)], which reads as $\frac{\text{Var}(J_\tau) \langle \sigma \rangle \tau}{\langle J_\tau \rangle^2 k_B} \geq 2$.

More details are added in the revised manuscript.

We have answered all the comments raised by the referees and we hope that the manuscript can now be found acceptable for publication in Nature Communications.

Reviewers' Comments:

Reviewer #1:

Remarks to the Author:

The authors have extensively revised the manuscript and addressed the comments that had been raised by the referees. This revised manuscript is ready for publication in Nature Communications in my opinion.

Reviewer #2:

Remarks to the Author:

The authors have made many changes to the manuscript and responded to the concerns of the reviewers. The resulting manuscript is improved.

There is one more conceptual clarification that must be addressed prior to a final decision on publication. It is related to the concerns raised by the other two reviewers who commented that the effective temperature is extremely large (130,000 K) as evaluated by the RMS mechanical displacement of the mode. The authors respond that the mode amplitude sets its effective temperature using standard equations and that it is indeed a thermodynamic temperature.

I believe a conceptual inaccuracy is arising related to this point. The temperature reported is indeed the (equivalent) temperature of the normal mode driven by white noise from the PZT. In fact, modes with frequency in the bandwidth of the white noise should all have similar temperature. However, there is another temperature one can define from a macroscopic perspective using the conductance of the membrane to a thermal bath (the wafer to which it is attached) and the heat flux dissipated in the membrane owing to the PZT driving. This driving is transferred to the other normal modes by anharmonic damping. The steady temperature is that which balances the incident heat flux with that removed by heat conduction (assume for now the membranes are decoupled and consider a single membrane). A calculation of this steady temperature is not necessary as the heat flux indicated in the y-axis of e.g. Fig 5a is extremely small and hence this steady temperature is essentially the bath temperature.

The point is that normal modes corresponding to atomic motions have the same displacements as in equilibrium as they are not driven by the white noise and the magnitude of the heat flux imparted by the white noise is miniscule. Normal modes within the bandwidth of the white noise, though, are heated to very high temperature and can participate in the heat transport process via the cavity mode coupling.

Thus the results presented here and the definition of temperature are to be interpreted as applying specifically to the relevant mechanical normal modes driven by white noise PZT. The temperatures are a convenient way to specify the RMS displacement amplitudes of the given normal modes and do not indicate the amplitudes of e.g. atomic normal modes of the crystalline solid which remain at their equilibrium values. The membrane therefore is characterized not by a single temperature due to the non-equilibrium.

I believe that this point needs to be made clearer, and some discussion of how the theory of non-equilibrium thermodynamics applies in this situation where various normal modes can be so out of equilibrium with each other is warranted. In its typical formulation non-equilibrium thermodynamics applies to two systems with different thermodynamic temperatures. Here, in a given membrane a subset of normal modes are out of equilibrium with the thermal bath; the other modes, e.g. those corresponding to atomic motions, remain unaffected by the heating. Heat transport arises via the coupling due to non-equilibrium of a subset of the normal modes.

Reviewer #3:

Remarks to the Author:

I have read the arguments of the authors and have not been convinced.

Response to the referee reports of manuscript NCOMMS-20-11938A by Yang et al:

In the following, we will respond to all comments from the referees and indicate changes made in the revised manuscript.

Response to the comments of the Reviewer 1:

(1) Comments: “The authors have extensively revised the manuscript and addressed the comments that had been raised by the referees. This revised manuscript is ready for publication in Nature Communications in my opinion.”

Response: We would like to thank the Reviewer 1 for recommending our manuscript to be published in Nature Communications.”

Response to the comments of the Reviewer 2:

(1) Comments: “The authors have made many changes to the manuscript and responded to the concerns of the reviewers. The resulting manuscript is improved.”

Response: We would like to thank the Reviewer 2 for this positive comment.

(2) Comments: “There is one more conceptual clarification that must be addressed prior to a final decision on publication. It is related to the concerns raised by the other two reviewers who commented that the effective temperature is extremely large (130,000 K) as evaluated by the RMS mechanical displacement of the mode. The authors respond that the mode amplitude sets its effective temperature using standard equations and that it is indeed a thermodynamic temperature.

I believe a conceptual inaccuracy is arising related to this point. The temperature reported is indeed the (equivalent) temperature of the normal mode driven by white noise from the PZT. In fact, modes with frequency in the bandwidth of the white noise should all have similar temperature. However, there is another temperature one can define from a macroscopic perspective using the conductance of the membrane to a thermal bath (the wafer to which it is attached) and the heat flux dissipated in the membrane owing to the PZT driving. This driving is

transferred to the other normal modes by anharmonic damping. The steady temperature is that which balances the incident heat flux with that removed by heat conduction (assume for now the membranes are decoupled and consider a single membrane). A calculation of this steady temperature is not necessary as the heat flux indicated in the y-axis of e.g. Fig 5a is extremely small and hence this steady temperature is essentially the bath temperature.

The point is that normal modes corresponding to atomic motions have the same displacements as in equilibrium as they are not driven by the white noise and the magnitude of the heat flux imparted by the white noise is miniscule. Normal modes within the bandwidth of the white noise, though, are heated to very high temperature and can participate in the heat transport process via the cavity mode coupling. Thus the results presented here and the definition of temperature are to be interpreted as applying specifically to the relevant mechanical normal modes driven by white noise PZT. The temperatures are a convenient way to specify the RMS displacement amplitudes of the given normal modes and do not indicate the amplitudes of e.g. atomic normal modes of the crystalline solid which remain at their equilibrium values. The membrane therefore is characterized not by a single temperature due to the non-equilibrium.”

Response: We agree with the referee’s comments. As the referee pointed out in his/her report, from a macroscopic viewpoint, a temperature could be defined by using the conductance of the membrane to a thermal bath and the heat flux dissipated in the membrane owing to the PZT driving. In the field of optomechanics, it is a well-accepted method to define the temperature by using the RMS displacement amplitudes of the given mechanical modes. For example, this definition of the temperature is adopted in the well-known ground-state cooling in optomechanics [Nature 475, 359 (2011), Nature 478, 89 (2011), Phys. Rev. Lett. 116, 063601 (2016)]. Here we also use this method to define the effective temperature of the mechanical normal mode driven by the white noise from the PZT. It is also true that the mechanical normal modes in the bandwidth of the white noise will be excited and have similar temperatures.

Following the referee’s suggestions, we have added the following discussion in the revised manuscript to make this point more clear. In the main text we added “Please note that the effective temperature is defined by the motion of mechanical oscillator, instead of the motions

of atoms which constitute the membrane.” In the Supplementary Information we added some details on how to define the temperature in the experiment. In addition, the following discussion is added in the revised supplementary: “It is worth mentioning that the temperature defined here is based on RMS displacement amplitudes of mechanical oscillators. The motions of atoms constituting the mechanical oscillators can lead to another definition of temperature, which remain at the equilibrium values with the environment. Therefore, the heat transport arises via the optomechanical coupling due to the two non-equilibrium mechanical modes at different thermodynamic temperatures.”

(3) Comments: “I believe that this point needs to be made clearer, and some discussion of how the theory of non-equilibrium thermodynamics applies in this situation where various normal modes can be so out of equilibrium with each other is warranted. In its typical formulation non-equilibrium thermodynamics applies to two systems with different thermodynamic temperatures. Here, in a given membrane a subset of normal modes are out of equilibrium with the thermal bath; the other modes, e.g. those corresponding to atomic motions, remain unaffected by the heating. Heat transport arises via the coupling due to non-equilibrium of a subset of the normal modes.”

Response: We have clarified the temperatures defined by the motions of mechanical oscillator and the motions of atoms in the revised manuscript. Since the membrane has various normal modes, then many normal modes can be out of equilibrium simultaneously. In the experiment, we choose (1,1) mode, which has a mode frequency ~ 400 kHz. The closest mechanical mode is (1,2) or (2,1) mode, which has a mode frequency ~ 645 kHz. Therefore, for the lowest few mechanical modes, they do not couple to each other, and they can be used as independent channels for heat transfer. We have demonstrated the point in the experiment through measuring their dynamics. The following discussions on how the theory of non-equilibrium thermodynamics applies in this situation are included in the revised manuscript: “Although one eigenmode of each membrane is utilized in this work, the multimode nature of the mechanical resonator implies the possibility of using this system as a multi-channel phonon-photon interface.” In conclusion, we agree with the referee’s comments that the heat transport studied

in our work is based on the optomechanical coupling of two spatially separated non-equilibrium mechanical normal modes. We thank the referee for helping to clarify the definition of the temperature, and make the manuscript more suitable for a broader readership.

Response to the comments of the Review 3:

(1) **Comments:** “I have read the arguments of the authors and have not been convinced.”

Response: The key point raised by the Reviewer 3 in the last round is the inconsistent of the theory and the experimental results. In the last response, we have pointed out that this discrepancy is due to the difference between the definitions of the instant and mean heat current. We also made the proper modifications in the manuscript to avoid this misunderstanding. We believe that we have well addressed all questions raised by the referee.

Reviewers' Comments:

Reviewer #2:

Remarks to the Author:

The authors have satisfactorily addressed my comments. I believe the article is now suitable for publication. My last comment is to encourage the authors to check the grammar and reconsidering their choice of words in certain cases: in particular, 'counterintuitively' in the abstract is perhaps not appropriate as at least two reviewers felt that the phenomenon was actually rather intuitive.

Response to the referee reports of manuscript NCOMMS-20-11938B by Yang et al:

In the following, we will respond to the comments from the Reviewer2 and indicate changes made in the revised manuscript.

Response to the comments of the Reviewer 2:

Comments: “The authors have satisfactorily addressed my comments. I believe the article is now suitable for publication. My last comment is to encourage the authors to check the grammar and reconsidering their choice of words in certain cases: in particular, 'counterintuitively' in the abstract is perhaps not appropriate as at least two reviewers felt that the phenomenon was actually rather intuitive.”

Response: We would like to thank the Reviewer 2 for recommending our manuscript to be published in Nature Communications. We have carefully checked the grammar. The word “counterintuitively” in the abstract has been deleted, and the whole sentence is modified to be “In the strong coupling regime, the instant heat flux spontaneously oscillates back and forth in the nonequilibrium steady states.”